Wheat gibberellin oxidase genes and their functions in regulating tillering

Wang Ting 1
Li Junchang 1
Jiang Yumei 1
Zhang Jing 1
Ni Yongjing 2
Zhang Peipei 1
Yao Ziping 1
Jiao Zhixin 1
Li Huijuan 1
Li Lei 1
Niu Yufan 1
Li Qiaoyun 1
Yin Guihong 1
Niu Jishan 1 jsniu@henau.edu.cn
1 Henan Technology Innovation Centre of Wheat/National Key Laboratory of Wheat and Maize Crop Science, Henan Agricultural University , Zhengzhou, Henan , China
2 Henan Engineering Research Center of Wheat Spring Freeze Injury Identification, Shangqiu Academy of Agricultural and Forestry Sciences, Shangqiu, Henan, China , Shangqiu , China
Hasanuzzaman Mirza
Electronic publication date: 2023 Sep 1
Publication date: 2023
Volume: 11
Electronic Location ID: e15924
Received 2022 Dec 28; Accepted 2023 Jul 30
Copyright: © 2023 Wang et al.
Copyright year: 2023
Copyright holder: Wang et al.
License: This is an open access article distributed under the terms of the Creative Commons Attribution License, which permits unrestricted use, distribution, reproduction and adaptation in any medium and for any purpose provided that it is properly attributed. For attribution, the original author(s), title, publication source (PeerJ) and either DOI or URL of the article must be cited.
License URL: https://creativecommons.org/licenses/by/4.0/

Keywords: Wheat (Triticum aestivum L.), Gibberellin oxidase (GAox), Tillering, Expression profiles, Gibberellin (GA), Gene transformation

Funding: National Natural Science foundation of China NSFC, 32171972 Internal Foundation of National Key Laboratory of Crop Science on Wheat and Maize SKL2022ZZ06 This study was supported by the National Natural Science foundation of China (NSFC, 32171972), and the Internal Foundation of National Key Laboratory of Crop Science on Wheat and Maize (SKL2022ZZ06). The funders had no role in study design, data collection and analysis, decision to publish, or preparation of the manuscript.

==============================
Multiple genetic factors control tillering, a key agronomy trait for wheat (Triticum aestivum L.) yield. Previously, we reported a dwarf-monoculm mutant (dmc) derived from wheat cultivar Guomai 301, and found that the contents of gibberellic acid 3 (GA3) in the tiller primordia of dmc were significantly higher. Transcriptome analysis indicated that some wheat gibberellin oxidase (TaGAox) genes TaGA20ox-A2, TaGA20ox-B2, TaGA3ox-A2, TaGA20ox-A4, TaGA2ox-A10 and TaGA2ox-B10 were differentially expressed in dmc. Therefore, this study systematically analyzed the roles of gibberellin oxidase genes during wheat tillering. A total of 63 TaGAox genes were identified by whole genome analysis. The TaGAoxs were clustered to four subfamilies, GA20oxs, GA2oxs, GA3oxs and GA7oxs, including seven subgroups based on their protein structures. The promoter regions of TaGAox genes contain a large number of cis-acting elements closely related to hormone, plant growth and development, light, and abiotic stress responses. Segmental duplication events played a major role in TaGAoxs expansion. Compared to Arabidopsis, the gene collinearity degrees of the GAoxs were significantly higher among wheat, rice and maize. TaGAox genes showed tissue-specific expression patterns. The expressions of TaGAox genes (TaGA20ox-B2, TaGA7ox-A1, TaGA2ox10 and TaGA3ox-A2) were significantly affected by exogenous GA3 applications, which also significantly promoted tillering of Guomai 301, but didn’t promote dmc. TaGA7ox-A1 overexpression transgenic wheat lines were obtained by Agrobacterium mediated transformation. Genomic PCR and first-generation sequencing demonstrated that the gene was integrated into the wheat genome. Association analysis of TaGA7ox-A1 expression level and tiller number per plant demonstrated that the tillering capacities of some TaGA7ox-A1 transgenic lines were increased. These data demonstrated that some TaGAoxs as well as GA signaling were involved in regulating wheat tillering, but the GA signaling pathway was disturbed in dmc. This study provided valuable clues for functional characterization of GAox genes in wheat.

Introduction

Gibberellins (GAs) comprise a large hormone family that modifies many aspects of plant growth and development (Spielmeyer et al., 2004), including seed germination (Yamaguchi & Kamiya, 2001; Urbanova & Leubner-Metzger, 2016), stem elongation (Tian et al., 2022), leaf expansion (Xu et al., 2016), flowering and fruit development (Yu et al., 2004). Currently, more than 136 GAs (Macmillan & Takahashi, 1968; Hedden, 2020) have been identified, and only GA1, GA3, GA4 and GA7 are recognized as major bioactive GAs in plants. Therefore, many non-bioactive GAs exist in plants as precursors for the bioactive forms or deactivated metabolites (Yamaguchi, 2008; Hedden & Phillips, 2000).

For more than a century, the biosynthesis and metabolic pathways of GAs have been studied in detail. GAs is biosynthesized from geranylgeranyl diphosphate (GGDP). Three different classes of enzymes are required for the biosynthesis of bioactive GAs from GGDP in plants: terpene synthases, cytochrome P450 monooxygenases, and 2-oxoglutarate-dependent dioxygenases (2ODDs) (Hedden & Thomas, 2012). Among them, three types of GA oxidases (GAoxs) in 2ODD gene superfamily are the key enzymes in the synthesis and degradation of GAs (Hu et al., 2021). They are mainly responsible for the mutual transformation of different GAs, and are particularly important in regulating the level of bioactive GAs. GA20-oxidases (GA20oxs) and GA3 Beta-hydroxylases (GA3oxs) are GA biosynthetic enzymes, they convert inactive GAs of GA12 and GA53 into bioactive GAs of GA1, GA3, GA4 and GA7 (Yamaguchi, 2008). In addition, it has been reported that GA7-oxidases (GA7oxs) are also involved in the biosynthesis of GAs. In pumpkin (Cucurbita maxima L.) and cucumber (Cucumis sativus L.), GA7-oxidases converted GA12-aldehyde to GA12 efficiently (Frisse, Pimenta & Lange, 2003; Lange et al., 2013). GA2-oxidases (GA2oxs) are recognized as GA deactivation enzymes, they convert GA precursors or bioactive GAs into inactive GAs by 2β-hydroxylation (Thomas, Phillips & Hedden, 1999; Kawai, Ono & Mizutani, 2014). GA1-oxidases (GA1oxs) are GA deactivation enzymes. In cucumber, GA1oxs convert GA9 to GA61 and GA4 to GA88, respectively (Lange et al., 2020). In wheat, GA1oxs convert GA9 to GA61 (Pearce et al., 2015).

So far, GAox genes have been identified in many plant species, including Arabidopsis (Arabidopsis thaliana L.) (Lange, Krämer & Lange, 2020), maize (Zea mays L.) (Ci et al., 2021), rice (Oryza sativa L.) (He et al., 2022), soybean (Glycine max L.) (Han & Zhu, 2011), jute (Corchorus capsularis L.) (Honi et al., 2020), tea plant (Camellia sinensis L.) (Pan et al., 2017), grape (Vitis vinifera L.) (He et al., 2019), and peach (Prunus davidiana L.) (Cheng et al., 2021). The functions of many GAox genes have also been elucidated. AtGA2ox9 contributes to freezing tolerance and AtGA2ox10 regulates seed production in Arabidopsis (Lange, Krämer & Lange, 2020). Studies of GA3ox mutants suggested that bioactive GAs synthesized in the stamens and/or flower receptacles were transported to petals to promote their growth in Arabidopsis (Hu et al., 2008). The semi-dwarf gene, sd1 of rice loses the function of GA20ox-2, leading to the sd1 mutant contained less GA levels than wild-type plants (Ashikari et al., 2002). OsHox4 gene was involved in GA metabolism, and controlled the expression of OsGA2ox and OsGA3ox family genes. Overexpression of OsHox4 gene caused bushy tillers (Zhou, Malabanan & Abrigo, 2015).

Tillering is an important trait of cereal crops that determines spike number, plant type structure, thereby affected the final crop yield (Wang et al., 2019). Tillering is a very complex trait, except the genetic factors and environmental factors, it can also be affected by plant hormones. Some hormones such as IAA (Cai et al., 2018), ABA (Liu & Hou, 2018; Lin et al., 2020), SLs (Nakamura et al., 2013) and CTK (Yang et al., 2020) are directly involved in regulating the tiller bud growth. GAs play promotable and/or inhibitory roles during shoot branching and tillering in different plant species. Several studies showed that GA treatment promoted axillary bud development and shoot branching in woody plants, such as Jatropha curcas, papaya (Chaenomeles sinensis), hybrid aspen and sweet cherry (Prunus avium) (Elfving, Visser & Henry, 2011; Ni et al., 2015a; Rinne et al., 2016). Some studies showed that GA affected tiller bud growth in annual herbaceous plants. In rice, the application of high concentration GAs could promote the degradation of rice OsSHR1 or MOC1, leading to fewer tillers (Lin et al., 2015, 2020). The inhibitory effects of GAs on tiller production were dose-dependent in tall fescue (Festuca arundinacea L.) (Zhuang et al., 2019). In bahiagrass (Paspalum notatum Flugge), overexpression of AtGA2ox1 resulted in a significant reduction of the endogenous bioactive GA1 contents, and increased the number of vegetative tillers (Agharkar et al., 2007). In tomato (Solanum lycopersicum L.), C19 GA2oxs silencing led to higher contents of active GA4 in axillary buds and few branches (Martínez-Bello, Moritz & López-Díaz, 2015). Overexpressing OsGA2oxs increased rice tiller number (Lo et al., 2008). In wheat, Rht24 encodes TaGA2ox-A9, which confers higher expression of TaGA2ox-A9 in stems, leading to a reduction of bioactive GAs in stems (Tian et al., 2022).

Previously, we have reported a dwarf-monoculm wheat mutant (dmc) derived from Guomai 301, and found the content of GA3 in the tiller primordia of dmc was significantly higher than that in Guomai 301. Transcriptome analysis revealed that GA biosynthetic genes (TaGA20ox-A2, TaGA20ox-B2 and TaGA3ox-B2) were lowly expressed, one GAs biosynthetic genes (TaGA20ox-A4) and two GAs catabolic genes (TaGA2ox-A10 and TaGA2ox-B10) were highly expressed in the tiller primordia of dmc (He et al., 2018; An et al., 2019). It indicates the existence of a negative feedback mechanism that regulates TaGAox genes expression. This study was to systematically analyze the roles of gibberellin oxidase genes played during wheat tillering. We identified 63 TaGAox genes in wheat genome, and comprehensive analyzed their gene and protein structures, and evolution. The expression profiles of TaGAox genes in tiller primordia of Guomai 301 and dmc after exogenous GA3 application were explored, and the effect of exogenous GA3 on wheat tillering was also evaluated. These results were reported here.

Materials and Methods

Plant materials and growth conditions

Guomai 301 (Wild type, WT) is a representative semi-winter wheat (Its vernalization requirement is between winter wheat and spring wheat; Vernalization at 0~12 °C for 15~35 days) cultivar in Henan, China. Mutant dmc was obtained from ethyl methyl sulfonate (EMS) treated Guomai 301 (Ni et al., 2015b). For GA3 treatment, seeds with full grains and the same size were selected, disinfected with 70% ethanol on the surface for 5 min, rinsed with distilled water, and placed in a petri dish for germination at 27 °C. After 48 h, the germinated seedlings were transplanted in experimental field, Henan Agricultural University, Zhengzhou, Henan Province, China (34°51′ N, 113°35′ E, 95 m a.s.l.). Each treatment had eight seedlings of Guomai 301 and eight seedlings of dmc, they were sown in the same plot, and each treatment had three replicates. For the hydroponics test, the germinated WT and dmc seeds were transplanted in tanks with 1.5 L of 1/2 Hoagland nutrient solution, and cultivated in an incubator. The seed germination conditions for genetic transformation were the same as above.

Identification of TaGAoxs

Wheat genome data and protein data were downloaded from the Ensembl Plants database (IWGSC refseqv1.1, http://plants.ensembl.org/index.html), and were used to identify all members of wheat GAox family. The Hidden Markov Model (HMM) profile of 2OG-FeII_Oxy (PF03171) and DIOX_N (PF14226) were downloaded from the Pfam database (http://pfam.xfam.org). HMMER software (http://www.ebi.ac.uk/Tools/hmmer) was used to search the TaGAox genes from wheat genome database. Meanwhile, the protein sequences of Arabidopsis GAox family members, maize GAox family members and rice GAox family members were downloaded from The Arabidopsis Information Resource (TAIR, http://www.Arabidopsis.org), Maize Genetics and Genomics Database (Maize GDB, http://www.maizegdb.org/gene_center/gene) and Rice Genome Annotation Project Database (RGAP, http://plantbiology.msu.edu), and these sequences were used as input sequences to BLASP in the wheat protein database. The output putative GAox protein sequences were confirmed by SMART (http://smart.embl-heidelberg.de) and Pfam searching for the presence of the 2OG-FeII_Oxy and DIOX_N domains. All output protein sequences with e-value ≤ 1e−10 were collected, removing the redundant sequences. The longest transcript sequence corresponding to each candidate gene was selected as the final sequence. Finally, obtained TaGAoxs were named mainly referred in Pearce et al. (2015).

The Expasy Prot Param tool (http://web.expasy.org/protparam) was used to predict the physical and chemical properties of wheat GAox proteins, including amino acid length, molecular weight, and theoretical isoelectric point and so on.

Construction of the phylogenetic tree

The multiple sequence alignment of wheat, Arabidopsis, maize and rice GAox protein amino acid sequences were carried out using MEGA software (http://www.megasoftware.net). Based on the sequence alignment results, the phylogenetic tree was constructed using Neighbor-joining method in MEGA software, setting the bootstrap parameter to 1,000 and using the default values for other parameters.

Analyses of conserved motif distributions and gene structures

The online software MEME (http://meme-suite.org) was used to analyze conserved motifs for each TaGAox protein sequences and the maximum number of motif finding was 10. The wheat Generic Feature Format 3 (GFF3) file was downloaded from the wheat genome database and used to elucidate the structure information of the TaGAox genes. Illustration depicting of protein motifs, conserved domains and gene structures of TaGAox genes was constructed using the TBtools software (Chen et al., 2020).

Analysis of the cis-acting elements

The 2,000 bp upstream sequences before transcription start positions of TaGAox genes were extracted from the wheat genome sequence, and the cis-acting elements were predicted and analyzed using the PlantCARE (http://bioinformatics.psb.ugent.be/webtools/plantcare/html/).

Gene duplication and synteny analysis of TaGAoxs

Multiple Collinearity Scan toolkit (MCScanX) was adopted to analyze the TaGAox gene duplication events, with the default parameters (Wang et al., 2012). Tandem duplication events were defined as chromosomal regions containing two or more genes within 200 kb (Holub, 2001). The synteny relationship of the orthologous GAox genes obtained from wheat, Arabidopsis, maize and rice was analyzed, and the syntenic maps were visualized by TBtools (http://github.com/CJ-Chen/TBtools). TBtools was used to calculate the synonymous rate (Ks) and nonsynonymous rate (Ka) substitutions of each duplicated gene pairs and their ratios (Ka/Ks) (Hurst, 2002).

Tissue specific expression analysis of TaGAoxs

The public wheat RNA-Seq datasets were downloaded from the Wheat Expression Browser (http://www.wheat-expression.com). It was used to analyze the expressions of TaGAox family genes in different tissues or organs (roots, stems, leaves, spikes and seeds) of Chinese Spring (Table S1). The gene expression level was represented by transcripts per million (TPM). The gene expression values present as log2-transformed normalized TPM values and visualized with TBtools.

GA3 treatment and RNA extraction

The seedlings at the three-leaf stage were treated with 2 × 10−4 mol/L GA3 to analyze the expression patterns of TaGAox genes. The tiller primordia of Guomai 301 and dmc at 0 h (untreated control), 1 and 2 h after GA3 application were sampled respectively. The RNAs of the samples were immediately extracted for gene expression analysis (Zhang et al., 2021).

qRT-PCR

qRT-PCR was performed as described previously (Zhang et al., 2021). The primers of TaGAoxs were designed using primer-blast of NCBI (www.ncbi.nlm.nih.gov/tools/primer-blast). All the primers were listed in Table S2. The Actin gene was used as an internal control. Each sample had three biological replicates. The relative expressions of TaGAoxs were calculated by 2−∆∆Ct methods (Livak & Schmittgen, 2001). All data were statistically analyzed. The values shown in the form of means ± SD were obtained from three independent experiments (Li et al., 2021).

Evaluation of the effects of GA3 on tillering

The wheat seedlings at the two-leaf stage were sprayed with distilled water and 2 × 10−4 mol/L GA3 solution respectively on the leaves until all the leaves were wet. Each seedling was sprayed with 5 mL solution, and treated once every 3 days for a total of 10 times. The tiller numbers were observed and counted once every 7 days from the sixth application, when the tiller number was obviously different among different treatments (Li et al., 2021).

Wheat transformation and expression analysis of TaGA7ox-A1

The CDS of TaGA7ox-A1 was isolated from Chinese Spring. TaGA7ox-A1 and plant expression vector of pCAMBIA1304 were digested with Nco I and Spe I. An expression vector pCAMBIA1304-CaMV35S:TaGA7ox-A1 was constructed by inserting TaGA7ox-A1 under of pCAMBIA1304-CaMV35S promoter, and transformed into Agrobacterium GV3101. pCAMBIA1304-CaMV35S:TaGA7ox-A1 was introduced into Guomai 301 by Agrobacterium mediated genetic transformation. When the seedlings of Guomai 301 grew to 0.5–1 cm, they were infected by Agrobacterium containing the expression vector. Agrobacterium infection solution was diluted to OD600 = 0.6 with 100 μmol/L of acetyl syringone. The transformed plants were transferred to a sterilized petri dish and cultured in a dark environment for 48 h. After 30 days of culture at 4 °C for vernalization, they were transplanted into an artificial climate incubator (Dong et al., 2018; Yang et al., 2021; Cao et al., 2021).

Specific primers F2/R2 were designed according to the sequences of the vector and the target gene (TaGA7ox-A1), and the positive TaGA7ox-A1-OE transgenic lines were detected by genomic PCR. The leaf DNAs of Guomai 301 and the transgenic plants were extracted at two-leaf stage and tillering stage. The amplified products were sequenced to demonstrate whether TaGA7ox-A1 gene was successfully transferred into Guomai 301. Under natural tillering conditions, tiller numbers of the transgenic and wild-type lines were observed and recorded every 7 days. The tiller primordia RNAs of Guomai 301 and the transgenic plants were extracted at tillering stage for transgene expression analysis. The primers used in gene isolation, transgenic plant verification and qRT-PCR were listed in Table S2.

Results

Identification of TaGAoxs

A total of 64 candidate wheat gibberellin oxidase (TaGAox) proteins were identified by HMMER and BLASTP in wheat Genome. The conserved motif search showed that one of the 64 proteins lacked the DIOX_N domain (TraesCS7D02G046300.1), the other 63 proteins were typical TaGAoxs family members (Table S3). The 63 TaGAox proteins included 13 TaGA20oxs, 3 TaGA7oxs, 6 TaGA3oxs, 40 TaGA2oxs and 1 TaGA1ox. The genes in the four TaGAox families were unevenly distributed on wheat chromosomes (Table S3).

The lengths of the amino acid sequences of TaGAoxs ranged from 305 aa (TaGA2ox-A5) to 442 aa (TaGA20ox-A2), with an average of 361.35 aa. The molecular weights of the TaGAoxs ranged from 33.30 to 48.13 KDa, with an average of 39.23 KDa. The TaGA20ox-A2 was the heaviest and TaGA2ox-A5 was the lightest. The predicted values of the isoelectric points of TaGAoxs ranged from 4.92 to 9.19. Among them, 14 TaGAoxs with theoretical pI greater than seven were slightly alkaline, while the remaining TaGAoxs with theoretical pI less than seven were slightly acidic. It was speculated that most of them were acidic proteins. Their aliphatic index ranged from 72.13 to 91.27, with an average of 80.97. For grand average of hydropathicity (GRAVY), TaGA2ox-A16 and TaGA2ox-D16 were two positive negative hydrophobic proteins, the other 61 were hydrophilic proteins (Table S3).

Phylogenetic tree of wheat GAox proteins

To investigate the phylogenetic relationships of TaGAox proteins, the phylogenetic tree was constructed using 127 GA oxidases of wheat, Arabidopsis, maize and rice (Fig. 1). Similar to previous report (Ci et al., 2021), the GAox proteins were clustered to seven distinct subgroups: C19-GA2ox (I, II), C20-GA2ox (III, IV), GA7ox (V), GA20ox (VI), and GA3ox (VII). They contain 19, 6, 12, 3, 3, 13 and 6 TaGAox proteins respectively. According to previous reports, TaGA1ox-B1 did not fit well into GA3ox subfamily, and TaGA1ox-B1 was shown to encode a GA1-oxidase (Pearce et al., 2015). The evolutionary relationship showed that the distribution of GA20oxs, GA3oxs and GA2oxs in each species was similar. There was no Arabidopsis GA7ox protein in GA7ox (V) subgroup. Most of TaGAox family members were clustered with maize and rice GAox family members. In summary, wheat GAox proteins were more similar to those of maize and rice than those of Arabidopsis.

Figure 1 The phylogenetic tree of the GAoxs in T. aestivum L. (Ta), A. thaliana (At), Z. mays (Zm) and O. sativa (Os).

Structures and conserved motifs of TaGAox genes

A phylogenetic tree of TaGAoxs was constructed using the 63 TaGAox protein sequences (Fig. 2A). The clusters of TaGAoxs were basically consistent with those in Fig. 1. To characterize the structures of TaGAox proteins, 10 motifs were identified using the MEME motif search tool (Table S4). The motif compositions and arrangements of the TaGAoxs in the seven subgroups were similar. Among them, motifs 1, 2, 3 and 7 were discovered in all of the TaGAox proteins. Motifs 3, 5 and 8 belonged to the DIOX_N domain, and motifs 1, 2, 4 and 6 belonged to the 2OG_FeII oxygenase domain (Fig. 2B). Motif 5 was unique to TaGA2ox subfamily. Motif 8 was found in GA3ox (VII) and GA20ox (VI) subgroups, and TaGA2ox-D11. Except for TaGA2ox-A5, motif 10 was found in five subgroups of C19-GA2ox (I, II), C20-GA2ox (III, IV) and GA20ox (VI). A more interesting thing was that motif 10 was located in N-terminal of C19-GA2ox (I, II) subgroups and in the C-terminal of C20-GA2ox (III, IV) subgroups. The exon-intron structure diagram of TaGAoxs showed that their genomic DNA sequence lengths were significantly different (Fig. 2C). The longest was TaGA2ox-B2, its length was about 7,500 bp, and the shortest was about 1,500 bp. The exon number of TaGAoxs was 1–5, most genes contained two or three exons. TaGA2ox5 only had one exon, TaGA3ox-D1 had five exons. As shown in the Fig. 2C, the homoeologous genes had more similar structures, such as TaGA3ox2 had three exons, which had little difference in structure, length and distribution and might have the same functions.

Figure 2 Phylogenetic relationships, conserved protein motif patterns and gene structures of TaGAoxs.

(A) The phylogenetic tree of TaGAox proteins. Clusters are indicated with different colors. (B) The motif compositions of TaGAoxs. The 1–10 motifs are displayed in different colored boxes, the scale at the bottom indicates the length of the amino acid sequences. The green dotted boxes represent DIOX_N domains and the brown dotted boxes represent 2OG_FeII oxygenase domains. (C) Exon-intron structures of TaGAoxs. Gray boxes indicate 5′- and 3′-untranslated regions; yellow boxes indicate exons; black lines indicate introns.

Cis-acting elements in the promoters of TaGAoxs

To gain a deeper understanding of the potential functions of the TaGAox family genes, we analyzed the cis-acting elements in the promoter regions of 51 TaGAox genes (the promoter sequences of the other 12 TaGAox genes contained a large number of ‘N’, so they hadn’t been analyzed) (Fig. 3). The results showed that there were a large number of cis-acting elements related to growth and development, abiotic stress and hormones in the promoter regions of the TaGAox genes. Among them, many cis-acting elements were involved in auxin response (TGA-element, AuxRR-core); abscisic acid response (ABRE), gibberellic response (P-box, GARE-motif, TATC-box), methyl jasmonate response (CGTCA-motif, TGACG-motif), salicylic acid response (TCA-element, SARE), light response (G-Box, Box 4, Sp1), and those growth-related cis-elements (CAT-box, GCN4_motif, O2-site, circadian, RY-element, CCAAT-box) and abiotic stress related elements (ARE, GC-motif, LTR, MBS, TC-rich repeats). The large numbers of elements were those of light-responsive elements, jasmonic acid-responsive elements, and ABA-responsive elements. Although all TaGAox genes encoding gibberellin oxidase, some of them did not contain gibberellin response elements, such as, TaGAox-D2, TaGA2ox-A1 and TaGA3ox-D1. The existence of various cis-acting elements in the gene promoter regions suggested that TaGAoxs played important roles in regulating wheat growth and development, stress response and hormone response.

Figure 3 The cis-acting elements in the promoters of TaGAoxs.

The shade of the blue represents the quantity.

Synteny relationships of TaGAoxs

During plant evolution, whole-genome duplications, transpositions, tandem gene duplications and segmental duplications play important roles in gene expansion and generation (Hu et al., 2021). In order to discover the gene duplication events of TaGAoxs, the 63 TaGAoxs were investigated. A total of 74 duplicated gene pairs were discovered, including 72 segmentally duplicated gene pairs, they distributed on different chromosomes (Fig. 4, Table S5). According to the methodology of Holub (Holub, 2001), there were two pairs of tandem duplication genes (Table S5). These results indicated that the expansions of TaGAox genes were mainly segmental duplications or tandem duplications, and the segmental duplication events played major roles in TaGAox genes evolution. To better determine the selective evolutionary pressure on TaGAox gene divergence, we calculated the ka/ks ratios of all the syntenic gene pairs. The ka/ks ratios of TaGA2ox4-1A and TaGA2ox4-1B-1 were 1.037, indicating that they had undergone neutral evolution, but the ka/ks ratios of the other 73 pairs of replicated genes were less than one (Table S6), indicating that the TaGAox genes mainly underwent purifying selection.

Figure 4 Schematic diagram of the chromosome distribution and inter chromosome relationships of TaGAoxs.

The gray lines indicate all duplicated gene pairs in wheat, the highlighted blue lines indicate probably duplicated TaGAox gene pairs.

In order to further infer the evolutionary relationships of the GAox genes, the comparative syntenic maps associated with wheat genome were constructed with other species, including Arabidopsis, rice, and maize. No syntenic gene of wheat GAox genes was found in Arabidopsis (Fig. 5A). A total of 30 TaGAox genes had syntenic genes in rice (Fig. 5B) and 34 TaGAox genes had syntenic genes in maize (Fig. 5C). The number of the orthologous gene pairs between wheat and rice, wheat and maize were 43 and 47, respectively. The ka/ks ratios of these gene pairs (Table S7) were calculated. All GAox gene pairs had ka/ks < 1, suggesting that TaGAox genes had undergone a strong purification selection pressure.

Figure 5 (A–C) Syntenic relationships of GAox genes between wheat and three representative species.

Gray lines in the background indicate the collinear blocks within wheat and other plant genomes, while the blue lines highlight the syntenic GAox gene pairs.

The expression patterns of TaGAoxs in different tissues

To gain insight into the putative functions of TaGAox genes, the expression profiles of TaGAoxs were analyzed using public RNA-Seq data from different organs/tissues of wheat (Fig. 6A, Table S1). Because it expressed at relative high levels in all tested tissues, TaGAox1 might play an important role during wheat development. TaGA2ox4, TaGA2ox3 and TaGA3ox2 were highly expressed in stem. TaGA20ox2 and TaGA2ox4 were highly expressed in spikes. Most of TaGAox genes were lowly expressed or not detected in the five tissues, implying functional redundancy of the TaGAox genes. The above results indicated that different TaGAox genes may be involved in different growth and development processes of wheat. The expressions of some genes were tissue-specific, which would contribute to different morphogenesis in plant development.

Figure 6 Expression profiles of TaGAoxs in various organs or tissues.

(A) Heatmap of expression profiles of TaGAoxs in various organs or tissues of Chinese Spring. (B) Heatmap of expression profiles of TaGAoxs in tiller primordia of Guomai 301 and dmc based on transcriptome data. Three biological replicates were set up in Guomai301 (T01, T02 and T03) and dmc (T04, T05 and T06). The gene expression values present as log2(FPKM). Note: Blue, Low expression level; Red, High expression level.

Based on our published transcriptome data (He et al., 2018), the expression profiles of all the TaGAox genes in Guomai 301 and dmc were further analyzed (Fig. 6B, Table S8). The expression profiles of TaGAoxs in tiller primordia showed that most genes expressed very lowly in all detected samples, these genes probably were not necessary for wheat tiller development. TaGA3ox-B2 and TaGA3ox-D2 were highly expressed in Guomai 301 and dmc. High expression levels suggested they played important roles during tiller development. Compared to WT, TaGA20ox-A4, TaGA2ox-A10 and TaGA2ox-B10 were highly expressed, and TaGA20ox-A2, TaGA20ox-B2 and TaGA3ox-A2 were lowly expressed in dmc. Differential expression of these genes might be one of the main causes containing tillering of dmc.

Expression patterns of TaGAoxs in response to GA3 application

In order to explore whether the expressions of TaGAox genes in wheat tiller primordia could be activated by GA3. Some TaGAox genes were selected to test according to their expression levels in dmc (Fig. 6B). It was found that GA3 significantly affected the expressions of TaGAoxs (Fig. 7). In dmc, the expressions of TaGA2ox genes were highest at 1 h after GA3 treatment. The expressions of TaGA3ox-B2 and TaGA3ox-D2 continuously decreased and those of TaGA7ox-A1 and TaGA7ox-D1 continuously increased. The expressions of TaGA20ox-B2 and TaGA3ox-A2 had no significant changes. The expressions of TaGA20ox-A4 and TaGA20ox-A1 were down-regulated at 2 h after GA3 treatment. The expression profiles of TaGAoxs in WT were different from those of dmc. The expressions of six TaGAoxs had no significant changes after GA3 treatment. The expressions of TaGA2ox-A1 and TaGA20ox-B2 decreased at 1 h after GA3 treatment, and recovered after 2 h. In both WT and dmc, TaGA7ox-D1, TaGA2ox10, and TaGA2ox-A16 were very sensitive to GA3 stimulation, the reason was considered as there were cis-acting elements of gibberellin response in their promoter regions. Although the gibberellin cis-acting elements of the TaGAoxs may be the same, most expression profiles of TaGAoxs in WT and dmc were significantly different.

Figure 7 Expression profiles of TaGAoxs in response to GA3 stimulation.

WT_0h, dmc_0h: untreated controls; WT_1h, dmc_1h: 1 h after GA3 treatments; WT_2h, dmc_2h: 2 h after GA3 treatments. Data were normalized to Actin gene and vertical bars indicated standard deviation. Asterisks indicate significant difference or highly significant difference between WT and dmc. An asterisk (*) and two asterisks (**) indicate significant difference (P < 0.05) and highly significant difference (P < 0.01) using Student’s t-test, respectively.

In summary, after GA3 application, the expressions of GAs catabolic genes and GA7ox genes were up-regulated, the expressions of GA biosynthetic genes were down-regulated.

Effects of GA3 on tiller formation of wheat

After the sixth time of GA3 application, the tiller number of Guomai 301 began to appear differences. Compared to the control, continuous GA3 treatment significantly increased the tiller number of Guomai 301, but didn’t affect tillering of dmc (Fig. 8). The results indicated that the exogenous GA3 could significantly promote tiller development of Guomai 301, but the dmc lacked response to GA3.

Figure 8 Tiller number changes of Guomai 301 and dmc in response to GA3 stimulation.

T1–T6: the GA3 treatment time points, the intervals were 7 days. An asterisk (*) indicate significant difference (P < 0.05) using Student’s t-test.

Function of TaGA7ox-A1 in regulating wheat tillering

Overexpression transgenic plants of TaGA7ox-A1 were obtained (Table S9). The positive transgenic plants were confirmed by PCR at early tillering stage and the amplified target fragments were sequenced at the late tillering stage (Fig. S1). These results indicated that the TaGA7ox-A1 was transferred into the positive transgenic plants (Fig. S2). Statistical analysis indicated that the average tiller numbers of TaGA7ox-A1-OE transgenic plants were significantly higher than those of WT from the TN4 stage. At the TN6 stage, the tiller number of wild-type plants was 6–7, while the tiller number of transgenic plants was 9–10 (Fig. 9A, Table S9). Compared to WT, the TaGA7ox-A1 gene relative expression levels and the tiller numbers of the transgenic plants increased significantly (Figs. 9A, 9B, Table S9). The wheat tiller number was significantly correlated with TaGA7ox-A1 gene expression level (r = 0.613, P < 0.05) (Table S9). This result indicated that TaGA7ox-A1 was involved in regulating wheat tillering.

Figure 9 The tiller numbers and qRT-PCR analysis of TaGA7ox-A1-OE transgenic lines.

(A) Average tiller numbers of the transgenic plants and controls (WT) at different stages. TN1–TN6: time points of the tiller number record, the intervals were 7 days. (B) TaGA7ox-A1 expression levels of the transgenic plants and WT. (C) Tiller numbers of the transgenic plants and WT (TN4 stage). An asterisk (*) indicate significant difference (P < 0.05) using Student’s t-test.

Discussion

Characteristics and evolution of wheat GAox gene family members

Gibberellins are important hormones in plants, and act during the whole life cycle of plants (Hernández-García, Briones-Moreno & Blázquez, 2021). In model plant Arabidopsis, the signaling pathways related to GAs have been well elucidated, and gibberellin oxidases are involved in the last step of GA biosynthesis pathway (Yamaguchi, 2006). However, the specific functions of the most GAox genes are largely remained unknown. Genome-wide predictions of GAox genes have become possible when many plants genomic sequences have been reported. In this study, we analyzed the structure, phylogenetic relationships, chromosomal locations, gene duplication events, cis-elements, and expression patterns of GAox genes in wheat.

A total of 63 TaGAox proteins with typical conserved domains, 2OG-FeII_Oxy and DIOX_N, had been identified in this study. Similar to other plant GAox proteins in Arabidopsis, rice, and soybean (Han & Zhu, 2011), the TaGAox proteins belonged to the 2OG-Fe (II) oxygenase superfamily. In this study, 13 TaGA20oxs, 40 TaGA2oxs, 6 TaGA3oxs, 3 TaGA7oxs and 1 TaGA1ox were identified, respectively. The preliminarily identified TaGAox genes included 10 TaGA20oxs, 29 TaGA2oxs, 6 TaGA3oxs and 1 TaGA1ox, and they were proved to have the corresponding oxidase activities (Pearce et al., 2015). Kumagai et al. (2022) identified 13 TaGA20oxs, this result was consistent with that of this study. Previous research identified the full-length genes of TaGA20ox2 and TaGA20ox3 in B and D genomes, but only partial sequences in A genome. In this study, the newly identified TaGA20ox-A2 and TaGA20ox-A3 were their homologous genes in A genome. The sequences of TaGA20ox-D3-2 and TaGA20ox-D3-1 were highly similar, and they were homologous genes. A study indicates OsGA20ox5 and OsGA20ox8 were clustered into GA7ox subfamily with CsGA7ox1 and CsGA7ox2 (Huang et al., 2015; Kawai, Ono & Mizutani, 2014; Sun et al., 2018). In this study, 3 TaGA7oxs and OsGA20ox5 and OsGA20ox8 are divided into subgroup V. So far, GA7ox activity was reported in pumpkin and cucumber, but has not been found in other species (Frisse, Pimenta & Lange, 2003; Lange et al., 2013). Therefore, whether the three TaGA7oxs have GA7ox activity needs to be further verified.

Compared to previous studies (Pearce et al., 2015), 11 TaGA2ox genes were newly identified in this study. Among them, TaGA2ox-A2, TaGA2ox-A7, TaGA2ox-A8 and TaGA2ox-D8 were homologous genes of TaGA2ox2, TaGA2ox7 and TaGA2ox8. Their homologous genes were used to prove that they have reduced substrate specificity for C19-GAs, as it effectively converted GA12 to GA110 as well as GA9 to GA51 (Pearce et al., 2015). It is speculated that these four newly identified TaGA2ox genes also have similar activities. In addition, TaGA2ox5 and AtGA2ox9-10 were divided into subgroup IV (C20-GA2ox) (Lange, Krämer & Lange, 2020). TaGA2ox15 and TaGA2ox16 and TaGA2ox8 were paralogous genes. Whether these newly identified TaGAox genes have the activity of GA2-oxidases oxidase needs further study.

Some reports indicated that the enzyme specificities of a few GAoxs were different from that of predicted by amino acid sequences, however, the enzyme activities were yet gibberellin oxidases (Frisse, Pimenta & Lange, 2003; Sun et al., 2018; Lange, Krämer & Lange, 2020; Lange & Lange, 2020). One of the reasons may be that some key nucleotide variations of a gene lead to amino acid sequence variations, subsequently the three-dimensional protein structure variations, which results in the changes of substrate specificity. Although it’s not completely accurate to functional predict the substrate specificities of gibberellin oxidases only by amino acid sequences, it’s the major methodology for studies of plant gene families, and are widely used in most studies (He et al., 2019; Ci et al., 2021; Du et al., 2022). Therefore, the classification of the TaGAox genes is reliable, though their substrate specificities need further research.

The 63 TaGAox genes were unevenly distributed on 21 wheat chromosomes. There were two pair tandem repeat genes and 72 pairs segmental duplications, which indicated that the gene duplications played an important role in the amplification of TaGAox genes (Xu et al., 2020), and there were functional redundancy among these genes (Panchy, Lehti-Shiu & Shiu, 2016). Phylogenetic tree analysis divided the TaGAox proteins into seven distinct subgroups (Fig. 1), which suggested the similarities of the gene structures and functions. Most of the TaGAox proteins clustered together with or close to OsGAoxs and ZmGAoxs and far away from AtGAoxs. This means that wheat, rice and maize are close relatives. There were no Arabidopsis GAox proteins in GA7ox (V) subgroup (Fig. 1). Apparently, some TaGAox genes evolved independently after differentiation of monocots and dicots. Gene structure analysis discovered that most TaGAox genes in the same subfamily had similar exon/intron structures, most of the genes contained two or three exons, and the result was similar to GAox genes in other species, such as Setaria italica, Sorghum bicolor, Hordeum vulgare, Brachypodium distachyon, maize, and rice (Zhang et al., 2022; Ci et al., 2021). Obviously, the structures of GAox genes in the same subgroup are conserved, which imply their similar biological functions.

The last step of bioactive GAs synthesis is catalyzed by GA20oxs and GA3oxs to convert GA12 and GA53 into active GAs (Yamaguchi, 2008), degradation of active C19-GAs and C20-GAs by GA2oxs through 2β-hydroxylation yields inactive GA products (Schomburg et al., 2003; Otani et al., 2010). Most common motifs of TaGAox proteins were shared by the seven GAox subgroups. However, some motifs of TaGAox proteins had been lost or added in the domains of 2OG-FeII oxygenase and DIOX_N, which might lead to gene functional changes (Fig. 2). For example, motif 4 were absent in TaGA7ox-D2, motif 5 was found in GA2ox. The motif 10 can distinguish the C19 GA2ox and C20 GA2ox subgroups. It was found that the conserved sequence LPWKET of GA20ox was located in motif 8 (Table S4) (Huang et al., 2015; Pan et al., 2017). Compared with TaGA20ox (VI), TaGA7ox (V) lacks motif 8, suggesting that it may have different oxidase activities. In summary, some unique motifs existed only in specific families. To better understand the functions of TaGAoxs, the biological functions of these special motifs need to be characterized further.

Signal transduction pathway of GAs is disturbed in dmc

GA signal transduction is a series of responses induced by cells after GA stimulation. Among them, GA-GID1-DELLA tricomplex plays an important role in the induction of plant growth and germination (Jiang & Fu, 2007). GA signal is perceived by GA receptor GIBBERELLIN INSENSITIVE DWARF1 (GID1), and regulate gene expression by promoting degradation of the transcriptional regulator DELLA proteins (Murase et al., 2008). In plants, when GA levels decreased, GA did not bind to GID1, so that DELLA gene inhibited the expression of GA responsive genes, thereby limited plant growth. When the GA level increased, GA and GID1 were combined to further GA-GID1-DELLA tricomplex. DELLA protein was hydrolyzed by ubiquitin-proteasome. The degradation of DELLA relieved the inhibition of GA response genes, and the plants showed normal GA response. DELLA can also induce the expression of upstream GA biosynthesis genes and GA receptors by feedback regulation (Wang & Deng, 2014). The Rht1 gene is a gain-of-function allele caused by an N-terminal truncation near the DELLA domain and Rht1 plants produce much more productive tillers (Kertesz, Flintham & Gale, 1991; Velu et al., 2017). Our studies had shown that DELLA and PIF3 were involved in GA signaling pathway and were highly expressed (Fig. S3). E3 ubiquitin-protein ligase were also lowly expressed in dmc (He et al., 2018; Li et al., 2016). It is speculated that the decrease of sensitivity to GAs is one of the factors constraining tillering in dmc.

Exogenous GA3 affects the expressions of TaGAox genes

Studies have shown that GA is one of the main hormones regulating stem elongation (Peng et al., 1999; Schomburg et al., 2003; Yamaguchi, 2008). The semidwarf rice sd1 is caused by the loss of function of the OsGA20ox2 gene. The precursor GA53 accumulates in the stems of sd1, the content of GA1 in sd1 is lower than that in tall lines (Spielmeyer, Ellis & Chandler, 2002). This study found that the expression levels of TaGA20ox-A2 and TaGA20ox-B2 in dmc were lower than that in Guomai 301 (Figs. 6, 7). Therefore, we speculated that homologous genes of TaGA20ox2 might be related to dwarfing of dmc. Plants can regulate GA balance by regulating expressions of GAox genes. In some crops, exogenous GA reduces the expressions of GA20oxs and GA3oxs, and increases the expressions of GA2oxs (He et al., 2019; Ci et al., 2021). The transcriptional analysis showed that the low expression levels of TaGA20ox-A2, TaGA20ox-B2, TaGA2ox-D7 and TaGA3ox-A2, and the high expression levels of TaGA20ox-A. TaGA2ox-10A and TaGA2ox-10B might be the feedback regulation of GA3 (Fig. 6B). qRT-PCR showed that GA3 treatment affected the expression of TaGAoxs. In dmc, TaGA2oxs were up-regulated after GA treatment, and GA2ox played a role in maintaining GA balance in GA synthesis and metabolism in plants (Thomas, Phillips & Hedden, 1999). TaGA3oxs were basically down-regulated by exogenous GA3, while TaGA20oxs showed different trends. Among them, TaGA7ox-D1, TaGA2ox10, TaGA2ox-A16 were most significantly regulated by GA3 (Fig. 7). TaGA7ox-D1 was highly expressed after GA3 treatment, but no GA responsive element was detected within its promoter. It may be that there are GA responsive elements outside its 2,000 bp promoter region, or there are other regulatory modes to promote the expression of TaGA7ox-D1. In summary, TaGA20ox-A2, TaGA20ox-B2, TaGA7ox1, TaGA2ox10, TaGA2ox-A16 and TaGA3ox-A2 may play a major role in regulating the level of bioactive GAs. But how do they regulate wheat tillering needs further research.

GA3 can promote wheat tillering

Studies indicate that GAs has a certain effect on plant tillering or branching. GAs negatively regulates tiller-related genes OsH1 and TB1 in rice, thus regulating the occurrence of tiller (Lo et al., 2008). In the tall fescue, GAs may inhibit tiller development by expressing FaTB1 in axillary buds (Zhuang et al., 2019). It is demonstrated that gibberellin synthesis inhibitor paclobutrazol (PBZ) can promote wheat tiller formation (Assuero et al., 2012). These results suggest that high levels of active GAs in plants inhibit branching or tillering. GA3 can inhibit the growth of tiller buds by controlling the content of IAA or cytokinin (CTK) in plants (Liu et al., 2011). Studies have shown that GA3 application can inhibit the growth of wheat tiller when wheat tiller buds begin to elongate (Cai et al., 2013). However, some researches have demonstrated that application of different concentrations of GA3 can increase the number of wheat tillers (Islam & Mehraj, 2014). Similarly, GA3 can stimulate tiller development in palmarosa (Cymbopogon martinii) (Khan et al., 2015). In this study, continuous application of GA3 from the second leaf stage could promote tillering of Guomai 301, but it had no obvious effect on tillering of dmc (Fig. 8). It was speculated that wheat at different stages had different responses to GAs. Our previous studies showed that the contents of GAs were significantly higher, and the contents of IAA were significantly lower in the tiller buds of dmc (An et al., 2019). These data demonstrated that GA signaling was involved in regulating wheat tillering, but the GA signaling pathway was disturbed in dmc.

TaGA7ox-A1 can promote wheat tillering

The plant overexpression vector of pCAMBIA1304-CaMV35S:TaGA7ox-A1 was constructed and transferred into Arabidopsis. The primary functional analysis showed that overexpression of TaGA7ox-A1 could significantly increase the branch numbers of the transgenic Arabidopsis plants (Fig. S4; Z. Jiao, 2020, unpublished data). Another study found that overexpression of OsGA20ox2 promoted plant height and tiller number (Qiao et al., 2013). In switchgrass (Panicum virgatum L.), overexpression of ZmGA20ox promoted tiller number (Do et al., 2016). In this study, overexpression of TaGA7ox-A1 significantly increased the tiller numbers of the transgenic wheat plants (Figs. 9, S2). The tiller number was positive correlated with the expression level of TaGA7ox-A1 (r = 0.613, P < 0.05). This result demonstrated that TaGA7ox-A1 could promote wheat tillering. However, the accurate molecular mechanism needs further study.

Conclusions

A total of 63 TaGAox genes distributed on 21 wheat chromosomes were identified. TaGAox genes belong to seven subgroups. The promoter regions of TaGAoxs contained a large number of cis-acting elements related to plant growth, hormone signaling pathway and stress response. Segmental duplication played a major role in TaGAoxs amplification. The TaGAox genes are tissue-specifically expressed. Genes of TaGA7ox-A1, TaGA20ox-A1, TaGA20ox-B1, TaGA2ox4 and TaGA3ox-2 played basic roles during wheat tillering. The abnormal expressions of TaGA20ox-A2, TaGA20ox-B2, TaGA3ox-A2, TaGA2ox10 and TaGA20ox-A4 were involved in the synthesis and metabolism of GAs in dmc tiller primordia, thereby affected tiller formation. The expressions of TaGAoxs were significantly affected by exogenous GA3. Exogenous GA3 significantly promoted tillering of Guomai 301, but the GA pathways were disturbed in dmc. Overexpression TaGA7ox-A1 promoted the transgenic wheat tillering.

Supplemental Information

Supplemental Information 1 Identification of TaGA7ox-A1-OE transgenic lines.

(A) Identification analysis of TaGA7ox-A1-OE transgenic lines by genomic PCR. (B) Genomic PCR multiple sequence alignment of TaGA7ox-A1-OE transgenic lines. M: Molecular weight marker 2000; WT: wild type Guomai 301; RP: recombinant plasmid containing TaGA7ox-A1. pC1304-TaGA7ox-A1: sequence of the recombinant plasmid containing TaGA7ox-A1. Red boxes: TaGA7ox-A1 gene sequence; Blue lines: vector sequence; OE-L349 and OE-L353: transgenic lines.

Click here for additional data file.

Supplemental Information 2 Phenotypes of the T0 TaGAox-A1-OE transgenic lines at different tillering stages.

A1-C1: Before tillering; A2-C2: tillering stage; A3-C3: After tillering; 1-21: TaGAox-A1-OE transgenic lines. WT1-3: wild type Guomai 301.

Click here for additional data file.

Supplemental Information 3 Plant hormone signal transduction (ko04075) in KEGG.

Red represents the genes with high expression levels in dmc, green represents the genes with low expression levels in dmc, and blue represents the genes with both low and high expression levels in dmc.

Click here for additional data file.

Supplemental Information 4 Phenotypes of the TaGA7ox-A1-OE transgenic Arabidopsis plants.

WT has only one fruit branch (A: white arrows) and no branch (B: blue arrows); the transgenic lines have 3–7 fruit branches (A: red arrows) and branches (B: red arrows).

Click here for additional data file.

Supplemental Information 5 The expression levels of TaGAox genes in different organs/tissues of Chinese Spring.

Click here for additional data file.

Supplemental Information 6 DNA sequences of the primers used in qRT-PCR.

Click here for additional data file.

Supplemental Information 7 The basic information of GAoxs in wheat.

Click here for additional data file.

Supplemental Information 8 The conserved motifs in wheat GAox proteins.

Click here for additional data file.

Supplemental Information 9 The duplication gene pairs of GAoxs in wheat genome.

Click here for additional data file.

Supplemental Information 10 One-to-one orthologous relationships of TaGAoxs.

Click here for additional data file.

Supplemental Information 11 One-to-one orthologous relationships of GAoxs between wheat and other species.

Click here for additional data file.

Supplemental Information 12 The expression levels of TaGAox genes in WT (T01, T02 and T03) and dmc (T04, T05 and T06).

Click here for additional data file.

Supplemental Information 13 Average of number tillers and qRT-PCR analysis of TaGA7ox-A1-OE overexpression lines.

Click here for additional data file.

Supplemental Information 14 The raw data for qPT-PCR.

Click here for additional data file.

Additional Information and Declarations

Competing Interests

Author Contributions

Data Availability

The authors declare that they have no competing interests.

Ting Wang conceived and designed the experiments, performed the experiments, analyzed the data, prepared figures and/or tables, and approved the final draft.

Junchang Li conceived and designed the experiments, performed the experiments, analyzed the data, prepared figures and/or tables, authored or reviewed drafts of the article, and approved the final draft.

Yumei Jiang conceived and designed the experiments, authored or reviewed drafts of the article, and approved the final draft.

Jing Zhang performed the experiments, prepared figures and/or tables, and approved the final draft.

Yongjing Ni conceived and designed the experiments, authored or reviewed drafts of the article, and approved the final draft.

Peipei Zhang performed the experiments, prepared figures and/or tables, and approved the final draft.

Ziping Yao performed the experiments, prepared figures and/or tables, and approved the final draft.

Zhixin Jiao analyzed the data, authored or reviewed drafts of the article, and approved the final draft.

Huijuan Li analyzed the data, prepared figures and/or tables, and approved the final draft.

Lei Li conceived and designed the experiments, authored or reviewed drafts of the article, and approved the final draft.

Yufan Niu analyzed the data, prepared figures and/or tables, and approved the final draft.

Qiaoyun Li conceived and designed the experiments, authored or reviewed drafts of the article, and approved the final draft.

Guihong Yin conceived and designed the experiments, authored or reviewed drafts of the article, and approved the final draft.

Jishan Niu conceived and designed the experiments, authored or reviewed drafts of the article, and approved the final draft.

The following information was supplied regarding data availability:

The raw measurements are available in the Supplemental Files.

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
