# Peer review of "Wheat gibberellin oxidase genes and their functions in regulating tillering"

_PeerJ, doi:10.7717/peerj.15924_

## Round 0.1 · original submission · Major Revisions

Please revise the manuscript based on the reviewers' comments.

Reviewer 1 ·

Basic reporting

The manuscript from Wang et al. describes an in silico approach to identify genes potentially involved in the gibberellin (GA) biosynthetic pathway in wheat and includes a description of their properties. In the second half of the manuscript, the authors use this information to study the expression levels of the different genes in an induced low-tillering mutant line discovered by the same group previously. Finally, the manuscript describes the overexpression of one of the identified genes and a brief description of tiller number in this line. The regulation of GA biosynthesis genes are important for a number of traits, including tillering, so there is value in a careful description of the different members of this family.
The in silico search for 2-ODD genes is similar to an earlier study that characterized GA20ox, GA3ox and GA2ox genes in wheat (Pearce et al. 2015 BMC Plant Biol. 15;130), which is not described in the current study. This earlier study was performed before a contiguous wheat genome assembly was available, so there is value in utilizing newly available genomic resources to provide a more complete characterization of each family. Because the authors did not see the earlier study, it is difficult to assess which novel genes from each family have been described, and which are the same genes already characterized. A revised manuscript should carefully describe the similarities and distinctions between their work and this previous study. It will be important to carefully consider gene names. Wherever possible, genes should be named consistently and logically, which were earlier based on orthologous rice and Arabidopsis gene names already described in the literature. The convention should be followed to refer to homoeologous genes – e.g. GA20ox-A1, GA20ox-B1, GA20ox-D1. It is not appropriate or necessary to include chromosome numbers in gene names and this should be avoided. Renaming genes that are already described extensively in the literature will lead to confusion for the community.
Each family is larger than the previous study, suggesting the existence of genes not previously described. The authors have assigned these novel genes to them to different families (GA20ox, GA2ox or GA3ox) based on sequence similarity, but it is important to note that these in silico predictions are insufficient to assign a gene to these families. Bioassays using different substrates (as described for existing gene members by Pearce et al. 2015) are required to accurately assign genes to these families and without this information which will be required to assign them to these families. Assigning them on this basis in a publication is misleading and should be approached with care. For example, bioassays performed by Pearce et al. demonstrated 1oxidase activity for a gene earlier predicted to encode a 3-oxidase. A quick look in the literature shows that the rice orthologs of two novel genes classified as “GA20oxidase” in the current study (OsGA20ox5 and OsGA20ox8) most likely exhibit GA7oxidase activity (e.g. https://pubmed.ncbi.nlm.nih.gov/30322023/ but also other literature that the authors should be aware of) and so should not be classified as GA20oxidases here. Therefore, any novel gene should be carefully described, and using in silico predictions based on sequence alone should be avoided. Describing gene families without any functional analysis is likely to be misleading for the community and should be approached carefully. It is most important that the authors consider first, which genes have already been described in the wheat literature (and use consistent naming) and secondly, for each novel gene, carefully describing the evidence that this gene does indeed encode a functional GA20oxidase, GA3oxidase or GA2oxidase enzyme. Where this is uncertain and the prediction is based simply on sequence similarity, I encourage the authors to use caution, to make sure that their results presented are well supported. Any reported data should be presented in context with the earlier study, describing what value this current analysis adds to the wheat research community.
In the second part of the study, the expression levels of the identified genes are analyzed in a dwarf monoculm (dmc) mutant that exhibits extremely low tillering and which was previously shown to have higher levels of GA. The authors observe that GA3 application increases the tillering rate in the wild-type genotype, but not the dmc mutant, and develop a transgenic line overexpressing one identified gene to observe the effects on tillering.
The finding that GA3 application increases tillering is inconsistent with decades of research from the literature across multiple species, including wheat. GA application is expected to destabilize DELLA which is associated with a reduced tiller number shown in multiple genotypes and chemical treatments, including the Green Revolution varieties that carry GA-insensitive forms of DELLA and the author’s own low-tillering mutant that has high levels of GA. Stating that GA application at different stages might have different effects is insufficient to describe such an unexpected result, since genotypes with stable reductions or increases in GA signaling always exhibit consistent effects on tillering. I would encourage the authors to double check this result and, if confirmed, carefully consider how this result could be explained. It seems possible that the dmc mutant is low tillering due to a block in GA signaling, which would account for the increase in the observed GA levels from the characteristic feedback regulation. The dmc mutant lines already have elevated GA levels, so the application of even more GA to these lines would not be expected to increase tillering.
From line 364 results of the transgenic experiment are described. The authors state that tiller number increased significantly, but no P-value is provided nor details of the statistical test. From the presented data in Figure 9, the tiller number in transgenic plants seems to be extremely variable and with overlapping error bars when considered overall (Figure 9A). Figure 9B is very difficult to interpret as different y-axes, but the absence of error bars in tiller number suggests that this was measured in a single plant for each transgenic line? Without additional replicates, further reporting or a broader characterization of these mutant lines, it is difficult to draw conclusions on the role of this gene. If included in the current study, this mutant lines should be characterized in detail, including in multiple replicates to provide confidence of the presented data.

Minor comments
Lines 42-43 – “Played a major role in tillering” – I advise caution in drawing conclusions on a gene’s role on the basis of expression levels alone.
Line 52, line 55 – “Were major causes of restraining tillering” – As above, a change in gene expression in a phenotype is just an association without any further experiments to test the involvement of the gene.
Lines 55-56 – Describing the role of GA on tillering in different genotypes can be clearer. “excluding dmc” is ambiguous.
Line 123 – The plant materials section is the best place to describe the different plants used in the experiment including the transgenic plants. This information is repeated in sections beginning line 185 and line 199.
Line 129 – Seeds were grown in a randomized design, but how many replicates were used in this field experiment?
Line 132 – It is important to include the version of the wheat genome used in this study. Which annotations and gene models were used?
Line 354-355 – “dmc is insensitive to GA3” – a lack of response to GA for tillering is not the same as being a GA-insensitive mutant and authors should be careful in the language used here.
Line 357 – “Transgenic plants were obtained” – A clearer description of the lines developed would be helpful for the reader here.
Line 358 – Replace “sequenceing” with “sequencing”.

Experimental design

See above.

Validity of the findings

See above.

Reviewer 2 ·

Basic reporting

1.In the Introduction part, it should be cited more references about the relationship between the GA and the tiller, such as “both SLs and GA will repress tiller buds growing” in reference: Nakamura H, Xue YL, Miyakawa T, Hou F, Qin HM, Fukui K, Shi X, Ito E, Ito S, Park SH, Miyauchi Y (2013) Molecular mechanism of strigolactone perception by DWARF14. Nat Commun 4(1):1–10; and “GAs usually inhibits stem branching, plants that overexpress the GA catabolism gene and GA-defcient mutants exhibit more branching phenotypes” in reference: Agharkar M, Lomba P, Altpeter F, Zhang H, Kenworthy K, Lange T (2007) Stable expression of AtGA2ox1 in a low-input turfgrass (Paspalum notatum Flugge) reduces bioactive gibberellin levels and improves turf quality under feld conditions. Plant Biotechnol J 5:791–801. It was more interested for readers.

Experimental design

1.For tissue specific expression analysis of TaGAoxs, you detemined the genes transcript levels in roots, stems, leaves, spikes and seeds. If the tiller primordia should be included, since they were sampled for the expression patterns of TaGAox genes after GA3 treatment.
2.Beyond of GA3 treatment, I suggest to use GA biosynthese inhibitor to further characterize the role of GA in tillering.

Validity of the findings

1.For the Discussion part, TaGA20ox5-4A can promote wheat tillering, it didn’t cited the related references to analyse the GA20ox gene function in tillering.

---

## Round 0.2 · Minor Revisions

Please revise the manuscript in accordance with the comments from Reviewer 1.

Reviewer 1 ·

Basic reporting

- The naming is much clearer. The authors should note the recent publication describing gene nomenclature in wheat, which are endorsed by the Wheat Initiative and can be a helpful guide for future reference (https://link.springer.com/article/10.1007/s00122-023-04253-w). I found the decision to name the three undefined genes as GAox confusing because this is the name given to all the 2ODDs in this study. These should be better defined when they are first introduced and could include a better description of what they are likely to be. If the other uncharacterized genes are named according to the similarity of genes with which they cluster, then wouldn’t it be appropriate to name these tentatively as GA7oxidases while making clear that they are remain to be functionally characterized? This is discussed on line 401.
- In the abstract (line 53) and elsewhere in the manuscript, the experiment to overexpress TaGAox in wheat should be better described. It is mentioned here as transferred, but overexpressing with a constitutive promoter is something different and should be made clear for the reader.
- Gene expression changes – Beginning on line 337 – The information on which genes are upregulated and downregulated should be summarized for the reader to better understand. Several GA2oxidase genes are upregulated, while biosynthetic genes are downregulated, the pattern that is commonly seen in tissues with very high levels of bioactive GA as part of the feedback mechanism and consistent with your results showing higher levels of bioactive GA and of no response to additional GA3 application (since this is already at saturating levels). From line 487 – I disagree with this conclusion that these genes play key roles in tillering. Their expression changes in ways that have been documented for decades as part of the feedback regulation of GA biosynthesis, which has somehow been disrupted in the GA-insensitive dmc mutant. Cloning the causative gene in the dmc mutant will reveal the reason, but the changes in expression identified in the current study are simply consequences of this and not evidence that they are involved in tillering. Likewise, the conclusion on lines 527-528 are not supported by your data.
Minor comments
Line 38 – Replace “Multiple genetic factors controlled tillering is a key agronomic trait…” with “Multiple genetic factors control tillering, a key agronomic trait…”
Line 43 – “were involved in regulating tillering”. Similar to my earlier comments, a change in expression is insufficient evidence to say a gene is involved in a trait, which instead requires characterization studies. Please replace this language to explain the result more carefully, that these genes were differentially expressed in the high-tillering dmc mutant.
Line 48 – Remove “The” from “The segmental duplication events…”
Line 54 – GAox1 was transferred into Guomai should be better described that this is overexpression, not complementation.
Line 82 – Replace “3b” with “3 Beta”.
Lines 126-128 – To help understand the basis of this study, it would be helpful to provide more context of these results. In which tissues of the dmc mutant were GA content higher? Which forms of GA? GA biosynthetic genes were differentially expressed, but which were upregulated and which downregulated, and were these changes consistent with the change in GA levels? These details are needed to help the reader understand the basis of the hypotheses behind the current study.
Line 138 – Please define “Semi-winter”.
Line 222 – Additional details are required of the transformation construct in order to understand this experiment. Which promoter was used to drive overexpression? How was expression assayed?
Line 238 – Replace “had not” with “lacked the”.
Line 470 – Replace “loses” with “loss”.
Figure 8 – no error bars, but this is mean tiller number. How many individuals and what was the variation?

Experimental design

See above.

Validity of the findings

See above.

---

## Round 0.3 · accepted · Accept

This version is properly revised.